# Intra-fused Gromov Wasserstein Discrepancy: A Smooth Metric for Cross-Domain structured Data

## Abstract

Optimal Transport (OT) theory, particularly the Wasserstein distance, is pivotal in comparing probability distributions and has significant applications in signal and image analysis. The Gromov-Wasserstein (GW) distance extends OT to structured data, effectively comparing different graph structures. This paper presents the Intra-fused Gromov-Wasserstein (IFGW) distance, a novel metric that combines the Wasserstein and Gromov-Wasserstein distances to capture both feature and structural information of graphs within a single optimal transport framework. We review related work on graph neural networks and existing transport-based metrics, highlighting their limitations. The IFGW distance aims to overcome these by providing an efficient, isometry-aware method for graph comparison that applies to tasks such as domain adaptation, word embedding, and graph classification, with applications in computer vision, natural language processing, and bioinformatics. We detail the mathematical foundation of IFGW and discuss optimization strategies for practical implementation.

## 1 Introduction

**Optimal Transport (OT)** Optimal transport theory (Villani et al., 2009), a branch of mathematics that studies the efficient transportation of resources between two distributions, has found increasing applications in various fields, including computer vision, machine learning, and natural language processing. In recent years, the theory has been extended to handle structured data, such as graphs, trees, and point clouds, enabling new and powerful techniques for data analysis and synthesis.

### 1.1 Related Work

Graph neural networks rely on training on structured data for various graph-related tasks. However, a common limitation is its difficulty of explaining the isomorphism in graph data. Identification of similarities between graphs is an essential problem in graph learning areas. In these areas, graph isomorphism problem is known as the exact graph matching (Xu et al., 2019b), which is not solvable in polynomial time nor to be NP-complete. Two graphs are considered *isomorphic* if there is a mapping between the nodes of the graphs that preserves their adjacencies. The graph isomorphism testing is used in a wide range of applications, such as the identification of chemical compound (Demetci et al., 2020), the generation of molecular graphs in chemical dataset (Titouan et al., 2019), and the electronic design automation (EDA) with placement and routing operations (Chan et al., 2000). Graph Isomorphism Network (GIN) (Xu et al., 2018) is recently proposed to implement Weisfeiler-Lehman (WL) graph isomorphism test (Shervashidze et al., 2011). However, such kind of approach only deals with graphs of same size, and hard to distinguish the difference between graphs with arbitrary sizes.

Notably, the optimal transport (OT) associated with their Gromov-Wasserstein (GW) discrepancy (Peyré et al., 2016), which extends the Gromov-Wasserstein distance (Mémoli, 2011), has emerged as an effective transportation distance between structured data, alleviating the incomparability issue between different structures by aligning the *intra*-relational geometries. GW discrepancy is isometric, meaning that the unchanged similarity under rotation, translation and permutation. Thanks to its favorable properties such as efficiency and isometry-awareness, GW has been extensively applied

to domain adaptation (Yan et al., 2018), word embedding (Alvarez-Melis & Jaakkola, 2018), graph classification (Vayer et al., 2018), metric alignment (Ezuz et al., 2017), generative modeling (Bunne et al., 2019), and graph matching and node embedding (Xu et al., 2019b;a; Xu, 2020).

The Fused Gromov-Wasserstein distance, a unified metric that interpolates between Wasserstein and Gromov-Wasserstein distances, offers a powerful approach for comparing graphs and structured data. By encoding both feature and structural information in a single OT formulation, it achieves strong performance in graph classification and clustering tasks. The Kantorovich formulation, which underlies the Wasserstein distance, emphasizes features while treating elements independently, whereas the Gromov-Wasserstein distance focuses on inter-element relationships, capturing structure but neglecting features.

Another method named CO-Optimal Transport (CO-OT) (Redko et al., 2020), is an innovative approach designed to address the limitations of traditional Optimal Transport (OT) frameworks, particularly in scenarios involving heterogeneous data spaces. COOT simultaneously optimizes transport maps for both samples and features, allowing for meaningful correspondences across different distributions without requiring a predefined cost function. However, despite its advantages, COOT faces certain shortcomings. One notable limitation is its computational complexity, which can become prohibitive in high-dimensional settings due to the need for managing large matrices and tensor operations. Additionally, while COOT provides a more interpretable mapping between datasets, its reliance on uniform weights can lead to suboptimal performance in cases where data distributions are imbalanced.

Table 1: Comparison of different methods based on structural and feature information, smoothness, and cross-domain capability.

| Graph Data | Structural Information | Feature Information | Smooth | Cross-domain |
|---|---|---|---|---|
| Wasserstein Villani et al. (2009) | X | X | ✓ | X |
| Unbalanced Wasserstein Liero et al. (2018) | X | X | ✓ | X |
| Gromov-Wasserstein Mémoli (2011) | ✓ | ✓ | X | X |
| Sample Gromov-Wasserstein Kerdoncuff et al. (2021) | ✓ | ✓ | X | X |
| Fused Gromov-Wasserstein Vayer et al. (2018) | ✓ | ✓ | ✓ | X |
| Our Proposed Methods | ✓ | ✓ | ✓ | ✓ |

To highlight, in this paper we proposed Intra-fused Gromov-Wasserstein (IFGW) distance combining the Wasserstein and Gromov-Wasserstein distances to capture both feature and structural information of graphs within a single optimal transport framework. Table 1 provides a comparison of different methods based on their handling of structural and feature information, smoothness, and cross-domain capability. The IFGW distance aims to overcome the limitations of existing transport-based metrics by providing an efficient, isometry-aware method for graph comparison that applies to tasks such as domain adaptation, word embedding, and graph classification. The proposed method has applications in computer vision, natural language processing, and bioinformatics, offering a versatile solution for comparing structured data across different domains.

## 2 INTRA-FUSED GROMOV-WASSERSTEIN DISCREPANCY

Before introducing our proposed IFGW, we first review the Wasserstein distance, Gromov-Wasserstein distance, and Fused Gromov-Wasserstein. Let $\Omega$ be an arbitrary Hilbert space, $D$ a metric on that space and $P(\Omega)$ be the set of Borel probability measures on $\Omega$. For any point $x \in \Omega$, $\delta_x$ is the Dirac unit mass on $x$.

**Definition 1** (Gromov-Wasserstein Distance). *Formally, Peyré et al. (2016) define the Gromov-Wasserstein (GW) distance between two measured similarity matrices $(\mathbf{C}, \boldsymbol{\mu}) \in \mathbb{R}^{n \times n} \times \sum_n$ and $(\mathbf{D}, \boldsymbol{\nu}) \in \mathbb{R}^{m \times m} \times \sum_m$ as follows*

$$\text{GW}(\mathbf{C}, \mathbf{D}) = \min_{\mathbf{T} \in \mathcal{C}_{\boldsymbol{\mu}, \boldsymbol{\nu}}} \sum_{i,j,k,l} \ell(\mathbf{C}_{i,j}, \mathbf{D}_{k,l}) \mathbf{T}_{i,k} \mathbf{T}_{j,l}, \tag{1}$$

where $\mathbf{C}$ and $\mathbf{D}$ are matrices representing structural metric between nodes within the graph. One common example is the *all-pair shortest path* (APSP) . $\ell(\cdot, \cdot)$ is the loss function either in square loss $\ell(a, b) = |a - b|^2$, or KL-divergence $\ell(a, b) = \text{KL}(a|b) = a \log(a/b) - a + b$ (Peyré et al., 2016).

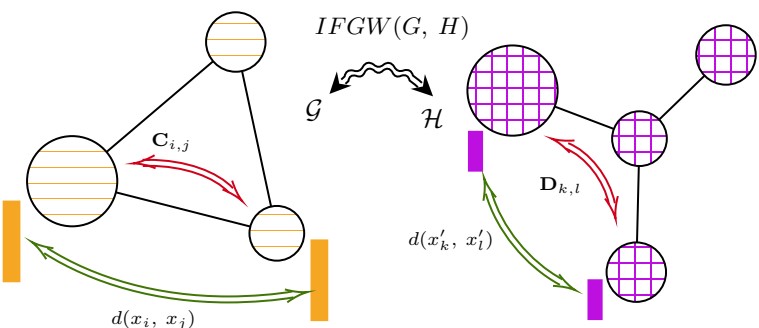

Figure 1: Illustration of IFGW, with two structured data from different domains. Domain $\mathcal{G}$ and $\mathcal{H}$ have two different types of nodes (with different dimension of features) and different structures. The IFGW distance aims to find the optimal transportation between the two domains by considering both feature and structural information.

Noting that KL divergence is a not a symmetric loss function, without loss of generality, we will keep using square loss in our following notations. $\{\boldsymbol{\mu} \in \mathbb{R}_n^+ : \sum_i \boldsymbol{\mu}_i = 1\}$ is the simplex of histograms with $n$ bins, which represents the node distribution. And $\mathbf{T}$ is the coupling between the two spaces on which the structural distance matrices are defined. Specifically,

$$\mathcal{C}_{\boldsymbol{\mu},\boldsymbol{\nu}} = \left\{\mathbf{T} \in \mathbb{R}_+^{n \times m}, \mathbf{T}\mathbf{1} = \boldsymbol{\mu}, \mathbf{T}^\top \mathbf{1} = \boldsymbol{\nu}\right\} = \mathcal{E} \cap \mathcal{N}. \tag{2}$$

Here, we denote the domain $\mathcal{E} := \left\{\mathbf{T} \in \mathbb{R}^{n \times m}, \mathbf{T}\mathbf{1} = \mathbf{p}, \mathbf{T}^\top \mathbf{1} = \mathbf{q}\right\}$ and $\mathcal{N} := \mathbb{R}_+^{n \times m}$. Basically, $\mathbf{T}$ is so-called doubly stochastic matrix. Furthermore, we can rewrite the problem in Koopmans-Beckmann form Koopmans & Beckmann (1957):

$$\mathsf{GW}(\mathbf{C}, \mathbf{D}) = \frac{\|\mathbf{C}\|_F^2}{n^2} + \frac{\|\mathbf{D}\|_F^2}{m^2} - 2 \max_{\mathbf{T} \in \mathcal{C}_{\boldsymbol{\mu},\boldsymbol{\nu}}} \mathrm{tr}(\mathbf{C}\mathbf{T}\mathbf{D}\mathbf{T}^\top). \tag{3}$$

Therefore, given the structural matrices $\mathbf{C}$ and $\mathbf{D}$, we primarily optimize with the trace of a quadratic form in (3) over the domain of $\mathcal{E} \cap \mathcal{N}$. GW distance is non-convex and highly related to the quadratic assignment problem (QAP), where it optimizes over the set of all permutation matrices. And QAP can be thought of finding the alignment of the nodes from two graphs that minimizes cost transfer from one to another, and it is a well-known NP-hard with no algorithm for solving this problem exactly in polynomial time. Basically, in the quadratic assignment problem, it optimizes over the domain of $\mathcal{E} \cap \mathcal{N} \cap \mathcal{O}$, where $\mathcal{O}$ is the orthonormal domain. Noting that $\mathcal{E} \cap \mathcal{N}$ is a convex hull of $\mathcal{E} \cap \mathcal{N} \cap \mathcal{O}$, indicating GW distance provides lower bounds of QAP.

**Optimization** The GW discrepancy problem can be solved iteratively by conditional gradient method (Peyré et al., 2016) and the proximal point algorithm (Xu et al., 2019a).

**Definition 2** (Fused Gromov-Wasserstein Distance). *Fused Gromov-Wasserstein (Titouan et al., 2019, FGW) distance defines the metric between structured object with the additional features from nodes. Formally, it can be written as:*

$$FGW(\mathbf{C}, \mathbf{D}, \mathbf{X}, \mathbf{X}') \tag{4}$$

$$= \min_{\mathbf{T} \in \mathcal{C}_{\boldsymbol{\mu},\boldsymbol{\nu}}} \langle (1-\alpha)\mathbf{M}_{\mathbf{X},\mathbf{X}'} + \alpha L(\mathbf{C}, \mathbf{D}) \otimes \mathbf{T}, \mathbf{T} \rangle \tag{5}$$

$$= \min_{\mathbf{T} \in \mathcal{C}_{\boldsymbol{\mu},\boldsymbol{\nu}}} (1-\alpha) \langle \mathbf{M}, \mathbf{T} \rangle_F + \alpha \sum_{i,j,k,l} (\mathbf{C}_{i,j} - \mathbf{D}_{k,l})^2 \mathbf{T}_{i,k} \mathbf{T}_{j,l} \tag{6}$$

$$= \min_{\mathbf{T} \in \mathcal{C}_{\boldsymbol{\mu},\boldsymbol{\nu}}} (1-\alpha) \langle \mathbf{M}, \mathbf{T} \rangle_F + \alpha \mathsf{GW}(\mathbf{C}, \mathbf{D}). \tag{7}$$

which can be simplified as $\min_{\mathbf{T} \in \mathcal{C}_{\boldsymbol{\mu},\boldsymbol{\nu}}} -2\,\mathrm{tr}(\mathbf{T}\mathbf{X}'\mathbf{X}^\top)$. The pairwise feature distance $\mathbf{M}_{\mathbf{X},\mathbf{X}'} = (d(\mathbf{X}_i, \mathbf{X}_j'))_{i,j}$ is a $n \times m$ matrix standing for the distance between the features. Normally, we have the Euclidean distance on as $d(\cdot, \cdot)$ to measure the similarity between features. FGW distance measures transportation cost from both structural and feature information by constructing a linear combination of GW distance along with a feature transportation.

**Intra-Fused Gromov-Wasserstein**     One key observation from FGW distance is that for the structural distance $\mathbf{C}$ measures the distance within the same graph. However, $\mathbf{M}$ measures the optimal distance cross different graphs. Therefore, it has the fundamental assumption that those two graphs need to have the same graph order, *i.e.*, $\mathbf{X}_i, \mathbf{X}'_j \in \mathbb{R}^h$. This is not always true in real scenarios, especially when the graphs have similar properties but with different types of nodes involved. To tackle this issue, we propose a new GW-based metric to measure optimal transportation considering both structural and feature information, named "Intra-Fused Gromov-Wasserstein" (IFGW) distance.

Let's revisit FGW distance first, it can be also rewrite as

$$\mathsf{FGW}(\mathbf{C}, \mathbf{D}, \mathbf{X}, \mathbf{X}') = \min_{\mathbf{T} \in \mathcal{C}_{\boldsymbol{\mu}, \boldsymbol{\nu}}} \sum_{i,j,k,l} \left[ (1-\alpha)d(\mathbf{X}_i, \mathbf{X}'_k) + \alpha(\mathbf{C}_{i,j} - \mathbf{D}_{k,l})^2 \right] \mathbf{T}_{i,k} \mathbf{T}_{j,l}. \quad (8)$$

Now, instead of taking the feature distance cross metric-measure spaces, denoted as $d(\mathbf{X}_i, \mathbf{X}'_k)$, we try to revise the Fused Gromov-Wasserstein distance based on intra feature distance. Specifically, we define Intra Fused-Gromov-Wasserstein Distance as:

$$\mathsf{IFGW}(\mathbf{C}, \mathbf{D}, \mathbf{X}, \mathbf{X}') = \quad (9)$$

$$\min_{\mathbf{T} \in \mathcal{C}_{\boldsymbol{\mu}, \boldsymbol{\nu}}} \sum_{i,j,k,l} \left\{ (1-\alpha)[d(\mathbf{X}_i, \mathbf{X}_j) - d(\mathbf{X}'_k, \mathbf{X}'_l)]^2 + \alpha(\mathbf{C}_{i,j} - \mathbf{D}_{k,l})^2 \right\} \mathbf{T}_{i,k} \mathbf{T}_{j,l}$$

$$= \min_{\mathbf{T} \in \mathcal{C}_{\boldsymbol{\mu}, \boldsymbol{\nu}}} \sum_{i,j,k,l} \left\{ (1-\alpha)\left(\mathbf{H}_{i,j} - \mathbf{H}'_{k,l}\right)^2 + \alpha(\mathbf{C}_{i,j} - \mathbf{D}_{k,l})^2 \right\} \mathbf{T}_{i,k} \mathbf{T}_{j,l}, \quad (10)$$

where we denote $\mathbf{H}_{i,j} := d(\mathbf{X}_i, \mathbf{X}_j)$ as the intra feature distance within the graph. Noting that in Eq (9), it takes the measure of both structural and feature internally, *i.e.*, $\mathbf{H}_{i,j}$ and $\mathbf{C}_{i,j}$. It's clear that this problem can be split into two GW distance formulations separately. However, those two GW distances need to be solved jointly with the same transportation matrix $\mathbf{T}$.

Therefore, we want to resort the optimization into the framework of vanilla Gromov Wasserstein. Considering the non-negativity of $(\mathbf{H}_{i,j} - \mathbf{H}'_{k,l})^2$ and $(\mathbf{C}_{i,j} - \mathbf{D}_{k,l})^2$, we will just split them apart so that no coupling term is involved.

Let's denote $\mathbf{D}_{i,j}(\alpha) = \alpha\mathbf{C}_{i,j} + (1-\alpha)\mathbf{H}_{i,j}$, where $\mathbf{C}_{i,j}$ and $\mathbf{H}_{i,j}$ represent the metrics on topology (shortest path) and feature ($\ell_2$-norm), respectively. Thus, we take the linear combination of both intra topological metric distance and intra feature metric. Overall, we have our final IFGW defined as

$$\mathsf{IFGW}_{\alpha}(\mathbf{C}, \mathbf{D}, \mathbf{X}, \mathbf{X}') \quad (11)$$

$$= \min_{\mathbf{T} \in \mathcal{C}_{\boldsymbol{\mu}, \boldsymbol{\nu}}} \sum_{i,j,k,l} \left[ \alpha\mathbf{C}_{i,j} + (1-\alpha)\mathbf{H}_{i,j} - \alpha\mathbf{D}_{k,l} - (1-\alpha)\mathbf{H}'_{k,l} \right]^2 \mathbf{T}_{i,k} \mathbf{T}_{j,l}$$

$$= \min_{\mathbf{T} \in \mathcal{C}_{\boldsymbol{\mu}, \boldsymbol{\nu}}} \sum_{i,j,k,l} (\mathbf{D}_{i,j} - \mathbf{D}'_{k,l})^2 \mathbf{T}_{i,k} \mathbf{T}_{j,l}. \quad (12)$$

From the Eq (11) we can see that it has the exact form of GW distance defined in Eq (1). And by setting $\alpha = 1$, IFGW distance induces to GW distance exactly. Figure 1 illustrates the metric-measure space of IFGW distance cross two different domains.

**Entropic Regularization of** $\mathsf{IFGW}$ **(Sinkhorn version).**     Due to the non-convexity of the $\mathsf{IFGW}$ distance, we propose to use the entropic regularization to approximate the solution. Considering the following entropic approximation of the vanilla $\mathsf{IFGW}$ formulation Eq (11)

$$\mathsf{IFGW}_{\epsilon}(\mathbf{C}, \mathbf{D}, \mathbf{X}, \mathbf{X}') \overset{def}{=} \min_{\mathbf{T} \in \mathcal{C}_{\boldsymbol{\mu}, \boldsymbol{\nu}}} \mathsf{IFGW}(\mathbf{C}, \mathbf{D}, \mathbf{X}, \mathbf{X}') - \epsilon H(\mathbf{T}), \quad (13)$$

where $H(\mathbf{T}) = -\sum_{i,j} \mathbf{T}_{i,j} \log \mathbf{T}_{i,j}$ is the entropy of $\mathbf{T}$, it is non-convex optimization problem, and we propose to use projected gradient descent, where both the gradient step and the projection are computed according to the KL metric. Iteration of this algorithm are given by

$$\mathbf{T} \leftarrow Proj_{\mathcal{C}_{\boldsymbol{\mu}, \boldsymbol{\nu}}}^{KL} \left( \mathbf{T} \odot e^{-\tau(\nabla \mathsf{IFGW}(\mathbf{C}, \mathbf{D}, \mathbf{X}, \mathbf{X}') - \epsilon \nabla H(\mathbf{T}))} \right), \quad (14)$$

where $\tau \geq 0$ is a small enough step size, and KL projection of any matrix $\mathbf{K}$ is

$$Proj_{\mathcal{C}_{\boldsymbol{\mu}, \boldsymbol{\nu}}^{KL}} \overset{def}{=} \arg\min_{\mathbf{T}' \in \mathcal{C}_{\boldsymbol{\mu}, \boldsymbol{\nu}}} KL(\mathbf{T}' \mid \mathbf{K}). \quad (15)$$

**Proposition 1.** *In the special case $\tau = 1/\epsilon$, iteration in Eq. (14) reads*

$$\mathbf{T} \leftarrow \mathcal{T}(\mathsf{IFGW}(\mathbf{C}, \mathbf{D}, \mathbf{X}, \mathbf{X}') \otimes \mathbf{T}, \boldsymbol{\mu}, \boldsymbol{\nu}). \tag{16}$$

*Proof.* Proof sketch is similar to Peyré et al. (2016). With the findings in Benamou et al. (2015), the project of a matrix can be understood as the solution to the regularized transport problem (Sinkhorn problem). Specifically, we can express the KL projection of any matrix $\mathbf{K}$ as

$$Proj_{KL}^{C_{\mathbf{p},\mathbf{q}}}(\mathbf{K}) = \mathbf{T}(-\epsilon \log(\mathbf{K}), \mathbf{p}, \mathbf{q}) \tag{17}$$

Additionally, we can derive the following equation:

$$\nabla \mathsf{IFGW}_{\epsilon,\mathbf{C},\mathbf{D}}(\mathbf{T}) - \epsilon \nabla H(\mathbf{T}) = \mathsf{IFGW}(\mathbf{C}, , \mathbf{D}) \otimes \mathbf{T} - \epsilon \mathbf{T}, \tag{18}$$

where we obtain the desired formula for the iteration when $\tau$ is specifically set to $1/\epsilon$. $\qquad \square$

**From Distance to Discrepancy** The GW distance is originally defined on the metric-measure space (mm-space), where the $\mathbf{C}$ is a strict metric from the structured data, *e.g.*, all-pair shortest path on graphs. However, we can replace the strict structural measure $\mathbf{C}$ with a pseudo-metric or semi-metric to generalize the GW distance to GW discrepancy. Examples like the adjacency matrix, Laplacian matrix and diffusion distance can be used in order to reduce the complexity of calculating the strict metric, while it still preserves the structural properties.

- GW is equivalent to IFGW with $\alpha = 1$, which is the same as FGW.
- FGW measures the feature distance cross mm-spaces.
- IFGW measures the feature distance within mm-space first, then takes the differences cross mm-spaces.

Given the $\alpha$ as an interpretation between the topological information and feature information, we can easily retrieve the lower bound of IFGW from an efficient algorithm Mémoli (2011) provides the FLB, SLB and TLB accordingly. (starting from definition 6.1 of Mémoli (2011)).

**IFGW Barycenters** We also define the Intra Fused Gromov-Wasserstein (IFGW) barycenters of measured similarity matrices $(\mathbf{C}_s)_{s=1}^S, (\mathbf{X}_s)_{s=1}^S$, where $\mathbf{C}_s \in \mathbb{R}^{N_s \times N_s}$ and $\mathbf{X}_s \in \mathbb{R}^{N_s \times H_s}$, using a Fréchet mean formulation:

$$\min_{\substack{\mathbf{C} \in \mathbb{R}^{N \times N} \\ \mathbf{X} \in \mathbb{R}^{N \times B}}} \sum_s \lambda_s \mathsf{IFGW}_\alpha(\mathbf{C}, \mathbf{C}_s, \mathbf{X}, \mathbf{X}_s). \tag{19}$$

However, noting that $\mathbf{D}$ is defined as the linear combination of structure distance $\mathbf{C}$ and feature distance $\mathbf{H}$. Therefore, we resort the problem into two parts. For the structure barycenter, we take benefits from Peyré et al. (2016, Prop. 4), where for the square loss, there is a closed form solution

$$\mathbf{C} \leftarrow \frac{1}{\mathbf{p}\mathbf{p}^\top} \sum_s \lambda_s \mathbf{T}_s^\top \mathbf{C}_s \mathbf{T}. \tag{20}$$

This basically indicates an evaluation after each optimal transportation obtained from IFGW. For the feature barycenter, we need to recover the proximal features so that the Eq. (19) is minimized. First, we reuse the trick in GW barycenter. Noting that $\mathbf{H}_s$ and $\mathbf{C}_s$ have the same dimension, therefore, instead of taking the structural similarity, we take intra feature similarity, *i.e.*,

$$\mathbf{H} \leftarrow \frac{1}{\mathbf{p}\mathbf{p}^\top} \sum_s \lambda_s \mathbf{T}_s^\top \mathbf{H}_s \mathbf{T}. \tag{21}$$

Second, we would need to find the feature barycenter by optimizing least square loss over

$$\min_{\mathbf{X} \in \mathbb{R}^{N \times B}} \|\mathbf{H} - d(\mathbf{X}, \mathbf{X})\|^2, \tag{22}$$

where $B$ is the dimension of feature in on barycenter, and it is not necessary to be one of the dimension in the original feature spaces. Due to the inherit of the convexity of the squared Euclidean distance, the optimization problem is convex.

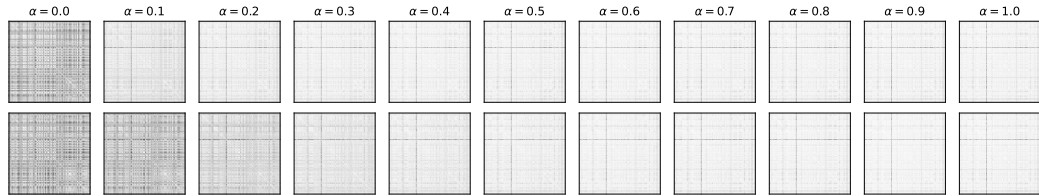

Figure 2: Comparison between FGW (first row) and IFGW (second row). Pairwise distance is calculated from MUTAG dataset, by setting different $\alpha$.

**Proposition 2.** *If the $\mathbf{C}_s$ and $\mathbf{H}_s$ are positive semidefinite (PSD) matrices, then the $\mathbf{C}$ and $\mathbf{H}$ corresponding to the barycenter are also PSD.*

*Proof.* Consider the derivative of IFGW given the cost distance combining the topological information and feature information, we have

$$\frac{\partial \mathsf{IFGW}}{\partial \mathbf{C}} = 2\left(\mathbf{D} - \mathbf{D}'\right) \odot \mathbf{TT}^\top. \tag{23}$$

Formulations Eq (20) and (21) shows that the update of $\mathbf{C}$ corresponds to a linear averaging of the matrices $diag(1/\mathbf{p})\mathbf{T}_s^\top \mathbf{C}_s \mathbf{T}_s$, which are all PSD since $\mathbf{C}_s$ are PSD and $\mathbf{T}_s$ are doubly stochastic. $\quad\square$

## 3 EXPERIMENTS

### 3.1 STRUCTURED DATA - GRAPH CLUSTERING

First and foremost, we utilize benchmark datasets to evaluate the clustering performance of the IFGW discrepancy, specifically focusing on structured graph datasets such as the MUTAG and QM9 for small molecules and PROTEINS and ENZYMES for bioinformatics. These datasets contain graphs of varying sizes and types, representing molecular structures, protein compounds, and enzymes with different relational properties. The graphs differ in both topological structures and node features, making them suitable for clustering tasks where both feature and structural information need to be considered.

Graph clustering aims to group similar graphs based on their structure and features, with applications ranging from bioinformatics (e.g., protein structure comparison) to chemistry (e.g., molecular compound classification). Traditional graph clustering methods, such as those based on graph edit distance or subgraph isomorphism, either suffer from computational inefficiency or ignore key aspects like feature similarity. Our goal is to demonstrate the effectiveness of IFGW in addressing these challenges by integrating both the structural and feature information into the clustering process.

Table 2 summarizes the clustering performance (mean accuracy scores with std.) of different methods on the benchmark datasets, including KMeans, spectral clustering, FGW, and IFGW. We evaluate the clustering results using two common metrics of features: normalized mutual information (NMI) and adjusted Rand index (ARI), serving as pairwise distance between feature matrices. Without explicitly tuning the hyperparameter, we take $\alpha = 0.5$ in both FGW and IFGW. To note that, KMeans and Spectral clustering are methods only consider the features, without accessing the geometric information of graphs. Our experiments show that IFGW outperforms FGW in clustering tasks across all evaluated datasets in general. Specifically, IFGW provides more nuanced clustering by considering both intra-graph feature distances and structural relationships, resulting in improved performance in scenarios where graphs have similar structures but different feature distributions, or vice versa.

**FGW v.s. IFGW** . To further analysis the rationals behind the experiment results, we also explore the reasons of the improved clustering performance of IFGW compared to FGW. Figure 2 illustrates the pairwise distance between graphs in the MUTAG dataset using FGW and IFGW with different $\alpha$ values. To note that when $\alpha = 1$, both FGW and IFGW degenerate to GW with same pair-wise distance matrix in the MUTAG dataset. We observe that FGW tends to sensitive to the feature

Table 2: Clustering performance of different methods on benchmark datasets.

| Dataset | Feature metric $d(\cdot, \cdot)$ | NMI | ARI | Cosine | Vector pair-wise |
|---|---|---|---|---|---|
| MUTAG | KMeans | $0.63 \pm 0.04$ | $0.43 \pm 0.05$ | $0.59 \pm 0.07$ | $0.41 \pm 0.06$ |
| | Spectral clustering | $0.68 \pm 0.03$ | $0.48 \pm 0.04$ | $0.64 \pm 0.05$ | $0.46 \pm 0.04$ |
| | FGW | $0.74 \pm 0.03$ | $\mathbf{0.75 \pm 0.02}$ | $0.71 \pm 0.04$ | $0.58 \pm 0.05$ |
| | IFGW | $\mathbf{0.81 \pm 0.02}$ | $0.71 \pm 0.03$ | $\mathbf{0.77 \pm 0.05}$ | $\mathbf{0.66 \pm 0.04}$ |
| QM9 | KMeans | $0.56 \pm 0.05$ | $0.39 \pm 0.06$ | $0.53 \pm 0.05$ | $0.38 \pm 0.05$ |
| | Spectral clustering | $0.61 \pm 0.04$ | $0.44 \pm 0.05$ | $0.57 \pm 0.06$ | $0.41 \pm 0.05$ |
| | FGW | $0.67 \pm 0.03$ | $0.52 \pm 0.04$ | $0.62 \pm 0.05$ | $\mathbf{0.69 \pm 0.02}$ |
| | IFGW | $\mathbf{0.72 \pm 0.03}$ | $\mathbf{0.60 \pm 0.03}$ | $\mathbf{0.68 \pm 0.04}$ | $0.65 \pm 0.03$ |
| PROTEINS | KMeans | $0.46 \pm 0.06$ | $0.42 \pm 0.07$ | $0.40 \pm 0.06$ | $0.30 \pm 0.06$ |
| | Spectral clustering | $0.49 \pm 0.05$ | $0.44 \pm 0.05$ | $0.44 \pm 0.06$ | $0.33 \pm 0.05$ |
| | FGW | $0.54 \pm 0.04$ | $\mathbf{0.51 \pm 0.04}$ | $0.49 \pm 0.04$ | $0.42 \pm 0.05$ |
| | IFGW | $\mathbf{0.60 \pm 0.05}$ | $0.47 \pm 0.04$ | $\mathbf{0.56 \pm 0.03}$ | $\mathbf{0.47 \pm 0.04}$ |
| ENZYMES | KMeans | $0.42 \pm 0.06$ | $0.27 \pm 0.07$ | $0.37 \pm 0.06$ | $0.43 \pm 0.07$ |
| | Spectral clustering | $0.47 \pm 0.04$ | $0.31 \pm 0.05$ | $0.42 \pm 0.06$ | $0.49 \pm 0.05$ |
| | FGW | $0.52 \pm 0.04$ | $0.38 \pm 0.05$ | $0.47 \pm 0.04$ | $0.46 \pm 0.04$ |
| | IFGW | $\mathbf{0.58 \pm 0.03}$ | $\mathbf{0.43 \pm 0.02}$ | $\mathbf{0.53 \pm 0.03}$ | $\mathbf{0.60 \pm 0.03}$ |

distances between graphs, leading significant difference once the feature information is introduced to the distance. In contrast, IFGW captures both feature and structural information in a balanced way, resulting in a more smooth clustering that considers both intra-graph feature distances and structural relationships. These findings suggest that the IFGW distance provides flexibility in clustering graphs with heterogeneous characteristics, making it a versatile tool for clustering tasks in domains such as bioinformatics and chemistry.

## 3.2 COMPUTER VISION - POINT CLOUD CLASSIFICATION

We also consider the tasks in computer vision, but with the data representation of point clouds. The MNIST and USPS datasets are widely used benchmarks for image classification tasks. The MNIST dataset [1] consists of 70,000 28x28 grayscale images of handwritten digits from 0 to 9. The USPS dataset [2], though similar in nature, contains 9,298 grayscale images, each of size 16x16, and serves as an alternative for evaluating machine learning models on digit recognition. Both datasets, originally consisting of rasterized images, were converted into point cloud representations for this experiment. A point cloud represents each image as a set of 2D points in pixel space, where only non-zero pixel values are retained. The goal is to test how a kernel-based classification model performs on point cloud data, which is inherently sparse compared to full image representations.

We introduce $\gamma$ to filter out the pixels with values greater or equal to $\gamma$, resulting a set of point clouds (examples are provided in Appendix). We follow the same setting as Nguyen & Tsuda (2023), converting a distance into a kernel matrix through the exponential function,, ie.e, $\mathbf{K} = exp((-\eta \mathbf{D}))$, where $\mathbf{D}$ is the pairwise distance (Euclidean distance between pixels) calculated based on IFGW. Due to the indirect computation of the kernel matrix, we can apply the kernel-based classification model, such as SVM, to classify the point clouds. Figure 3 shows the classification results of SVM on the MNIST and USPS datasets using IFGW with different $\gamma$ values (threshold). The results suggest that with more pixels involved in the graph, the classification performance improves, indicating that the IFGW distance can effectively handle sparse data and capture the underlying structure of point clouds.

---

[1]MNIST point cloud: https://www.kaggle.com/datasets/cristiangarcia/pointcloudmnist2d

[2]USPS point cloud: https://www.kaggle.com/datasets/bistaumanga/usps-dataset

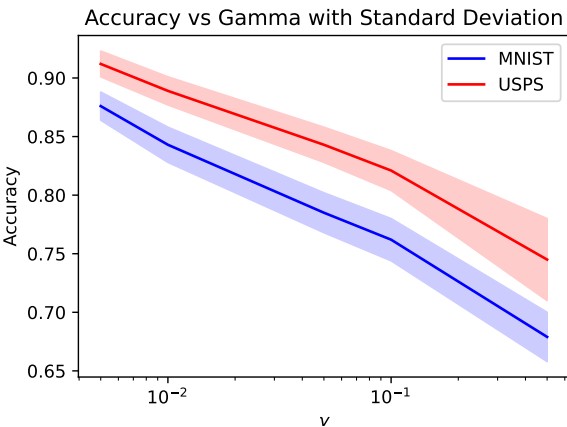

Figure 3: Point cloud classification on MNIST and USPS datasets (SVM).

### 3.3 GRAPH SIMILARITY SEARCH: APPLICATIONS AND CHALLENGES

Graph similarity search has numerous applications across various domains, including image retrieval, document retrieval, protein structure comparison, and chemical compound discovery. More specifically, scientists can use graph similarity to identify proteins with similar structures, which may have similar functions and serve as potential drug targets in protein structure comparison. For chemical compound discovery, chemists can use graph similarity to identify compounds with similar structures and potentially analogous properties, which may lead to similar applications or candidates for further research.

SMILES (Simplified Molecular Input Line Entry System), a widely used representation of chemical compounds, converts molecular structures into graph form for comparison. However, traditional SMILES-based comparisons rely on Jaccard similarity, which focuses solely on structural similarity without considering atom or bond properties. This approach, while useful, can overlook critical aspects of chemical behavior determined by node features (e.g., atom types) and edge properties (e.g., bond types).

Alternatively, the Intra-Fused Gromov-Wasserstein (IFGW) distance offers a more holistic approach by combining structural and feature-based comparisons in a single framework. Unlike methods that focus solely on structure, IFGW integrates the node features, making it suitable for more complex applications requiring a comprehensive similarity metric. Additionally, IFGW is isometry-aware, which means it accounts for the intrinsic geometric properties of graphs. This feature is particularly valuable for applications such as protein structure comparison and chemical compound discovery, where both the geometry and chemical properties of molecules or proteins influence their function.

**Cross-domain similarity.** To demonstrate, consider the structure of (-)-L-Carnitine (SMILES: C[N+](C)(C)CC(CC(=O)[O-])O), which exists as one of two stereoisomers: the enantiomers D-carnitine (S-(+)-) and L-carnitine (R-(-)-). Both are biologically active, but only L-carnitine naturally occurs in animals, while D-carnitine is toxic as it inhibits the activity of the L-form. We extracted both the 2D structure and 3D conformer of L-carnitine from the PubChem database and converted them into graphs. In these graphs, the nodes represent the atoms (excluding hydrogens), and the features include either 2D or 3D coordinates, along with atom type information. Using IFGW, we compared the 2D and 3D graphs of L-carnitine to measure the similarity between different domain representations of the same structure. The results showed that IFGW effectively captured both structural and feature differences between the 2D and 3D graphs, yielding a dissimilarity score of 0.0013 for the same molecule. Notably, we did not adjust the hyperparameter $\alpha$, leaving it at 0.5. The lower the score, the greater the similarity, indicating that the 2D and 3D graphs are highly similar, even without prior knowledge of the structure.

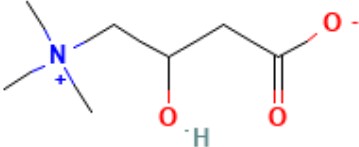
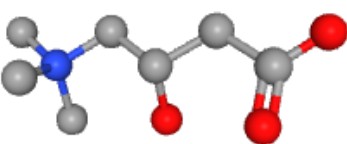

(a) 2D Structure of L-Carnitine                    (b) 3D Conformer of L-Carnitine

Figure 4: Comparison of 2D and 3D structures of L-Carnitine under IFGW distance is 0.0013.

## 4    CONCLUSION

This paper introduces the Intra-Fused Gromov-Wasserstein (IFGW) distance, a novel metric combining the strengths of Wasserstein and Gromov-Wasserstein distances to compare both structural and feature information of graphs. IFGW's flexibility and isometry-awareness make it a powerful tool for cross-domain structured data comparisons, offering substantial improvements over existing methods. In our experiments, IFGW demonstrated its versatility across tasks such as graph clustering, point cloud classification, and graph similarity search. For graph clustering, it consistently outperformed other methods by integrating feature and structural information, particularly in bioinformatics datasets like MUTAG and QM9. In point cloud classification of MNIST and USPS datasets, IFGW handled sparse data effectively, showcasing its potential for image-based tasks.

Crucially, IFGW can be extended to more complex domains like 3D molecular structure comparison, where 3D coordinates serve as features. This capability is especially valuable for tasks such as protein structure analysis, where both the spatial arrangement of atoms and their relationships are critical. Additionally, IFGW's framework is adaptable to mRNA sequential data, where the sequential order of nucleotides can be treated as a graph feature, making it useful for genomic applications. The metric of structured data could also be used in contrastive learning, where the distance between data points is minimized in the same class and maximized in different classes. We will leave this as future work.

### 4.1    LIMITATIONS

While IFGW offers a robust and flexible approach to graph similarity search, several challenges remain:

**Computational Complexity**: Traditional approaches like subgraph isomorphism and graph edit distance are computationally expensive, especially for large graphs or databases. IFGW, though more efficient, still requires optimization to handle very large datasets.

**Scalability**: As datasets grow in size, efficiently searching large databases of graphs becomes increasingly challenging. Methods that rely on pairwise comparisons, such as graph edit distance, often struggle with scalability. IFGW helps mitigate this by offering a smooth distance function that facilitates faster computation .

**Data Heterogeneity**: Graphs can vary in size, structure, and feature types. A universal similarity metric must be flexible enough to handle this heterogeneity. IFGW addresses this by incorporating both structural and feature information in a single framework, however, validating the success of such metric still domain experts to be involved intensively.

**Interpretability**: Some graph similarity metrics, including IFGW, can be difficult to interpret, making it challenging to understand why two graphs are considered similar or dissimilar. Future work could focus on improving the interpretability of these metrics.

**Robustness**: Graph similarity metrics may be sensitive to noise, outliers, or missing data, which can lead to inaccurate or unstable results. Ensuring robustness in graph similarity search remains an open research question .

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

## A    APPENDIX

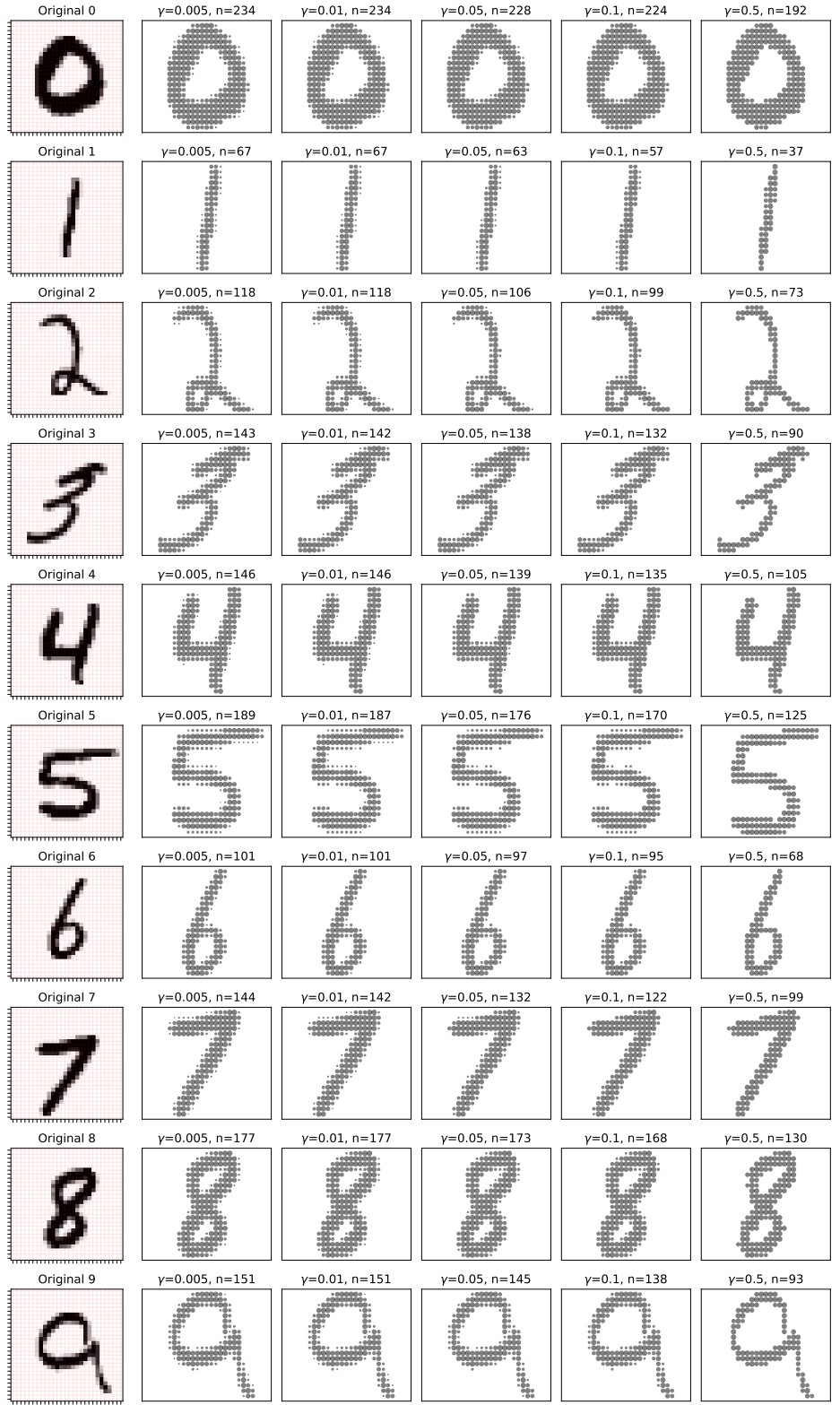

Figure 5: MNIST point cloud with threshold filter

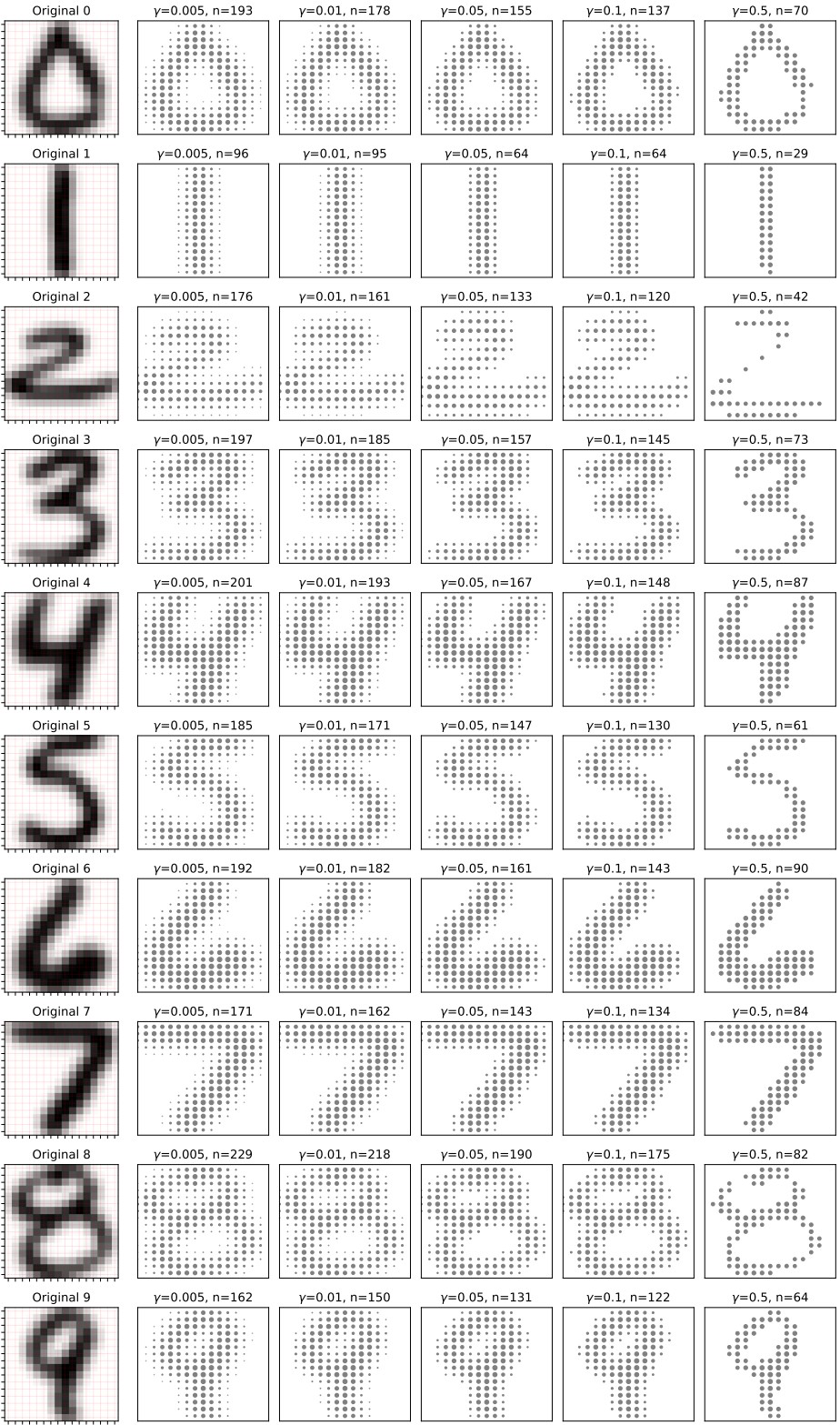

Figure 6: USPS point cloud with threshold filter

