# OpenReview forum: "Intra-fused Gromov Wasserstein Discrepancy: A Smooth Metric for Cross-Domain structured Data"
_ICLR.cc/2025/Conference — ICLR 2025 Conference Withdrawn Submission_

### Official Review · Reviewer_J75b · 2024-10-25

**Soundness:** 1
**Presentation:** 3
**Contribution:** 1
**Rating:** 1
**Confidence:** 5

**Summary:**

Paper defines the "intra-fused Gromov-Wasserstein" (IFGW) discrepancy and claims that it combines a Wasserstein and a Gromov-Wasserstein distance within a single framework.
In more details, within the FGW formulation, it replaces the matrix computed between the features $d(x_i, x'_k)$  by a (squared) difference between the intra-domain features $d(x_i, x_j) - d(x'_k, x'_l)$.

An entropic regularization is then added to IFGW to ease its approximation.
A set of experiments is provided in several scenarios: graph clustering, point cloud classification and graph similarity.

**Strengths:**

- The problem that is tackled is important and difficult
- The proposed IFGW formulation is clear and builds on a state-of-the-art distance

**Weaknesses:**

My main concern is the formulation of IFGW.
Firstly, it is unclear to me why the intra-domain distance matrices $C$ and $D$ cannot include values like $d(x_i, x_j)$ and $d(x'_k, x'_l)$. In the original FGW paper, a shortest path was used, but any other metric could be considered.
It seems (although the interpretation is unclear, as matrix $D$ refers to two distinct quantities with the same notation) that IFGW essentially reduces to a GW problem with cost matrices that involve two (weighted) cost functions (see eq. 12).
Moreover, the transition from eq. (10) to eq. (11) appears to be incorrect.

Other comments:
- There are several typos throughout the text.
- Table 1: I am uncertain about the meaning of “smooth” and “cross-domain,” as GW is reported as not being applicable in this context. I am also unclear as to why UOT is included in this table.
- After eq. (2): Is the method restricted to uniform masses (since it is stated that $T$ is a doubly stochastic matrix)?
- Please verify eq. (3) and its relevance to the paper.
- First line after eq. (7) and its simplification: could you provide more details on this claim?
- Experiments: Why was clustering chosen over a classification task in Section 3.1? Why was $\alpha = 0.5$ set rather than cross-validating it? Section 3.2 should clarify the experimental setup and conclusions (for instance, what does $\nu$ represent in Figure 3?). In Section 3.3, it is challenging to draw conclusions from the visual inspection of only two molecules.

**Questions:**

- what is the difference between IFGW and a vanilla GW with a dedicated cost matrix?

---

### Official Review · Reviewer_NX9a · 2024-10-29

**Soundness:** 2
**Presentation:** 2
**Contribution:** 1
**Rating:** 1
**Confidence:** 5

**Summary:**

This paper presents an extension of the Gromov-Wasserstein (GW) distance for graph that deals with multi-modal data on nodes. The propose formulation allows to directly extends most current results on the GW distance. Thus, the computation of the GW barycenters is straightforward. The experiment shows interesting results compare to some state-of-art methods.

**Strengths:**

The proposed framework is well described and motivated. Cross-domain structured data are know to be difficult to managed, especially when the modalities are too different. The idea of the paper tackle this issue by building an appropriate metric (using a convex combination of metrics) and insert it into a GW distance. Thus, it offer all the power of the GW framework which is a, now, well studied distance with effective methods for computing a solution.

**Weaknesses:**

This paper shows substantial weakness that make irrelevant for a publication. I split my remarks into major (the ones that really block the publication) and minor (important but non-blocking).

## Major remarks

1. **Novelty**: the main contribution of this article can be sum up as using GW distance with a compound metric on cross-domain data. It is not a new result about GW distance or a new variant of the optimal transport framework. The compound metric is an old idea that can be found in many other papers dealing with graph (see [1] for example). Furthermore, none of the proposition or theorem are new, they are directly taken from [2]. Secondly, the Fused-Gromov-Wassertein allows cross-modal domain data on both nodes and edges, it is just a matter of metrics.

2. **Writing**: while reading the paper, it seems incomplete as if sentences were missing. For example between the first paragraph of section 1.1 and the second paragraph, there is no transition. The text jumps from graph neural networks to optimal transport without any details on how these two frameworks can be related. Furthermore, the presentation of the GW distance is incomplete with notations that appear from nowhere. Another example, the first sentence of page 3 is with no link with the second sentence...

3. **Experiments**: the experiments are incomplete, it need more comparison against state-of-art methods. Since we are dealing with graph, I would expect comparison against FGW [3], KerGM [4] or GWL [5]. Since we have a distance classification results are also possible.

## Minor remarks

1. It seems there is a confusion between graph isomorphism and graph matching problems. While the complexity class of the first is still research question, the second is known to be NP-Complete. Since the GW distance is trying to solve the graph matching problem, it would be more appropriate to have a discussion about it.

2. In page 3, the authors formulation the GW distance as a Koopmans-Beckmann QAP. However, it the first time in the paper that the QAP appears. I think the QAP Koopmans-Beckmann should be presented in the related works section.

3. I agree that the GW distance is not convex, but the problem presented in (3) is convex... Seems there is a confusion between the set all permutation matrices (non-convex set) and the Birkhoff polytope (convex).

4. In page 4. I don't understand the notion of graph order. Does it means the dimension of features is the same?

## Reference

[1] Mahé, P., Ueda, N., Akutsu, T., Perret, J. L., & Vert, J. P. (2004, July). Extensions of marginalized graph kernels. In Proceedings of the twenty-first international conference on Machine learning (p. 70).
[2] Peyré, G., Cuturi, M., & Solomon, J. (2016, June). Gromov-wasserstein averaging of kernel and distance matrices. In International conference on machine learning (pp. 2664-2672). PMLR.
[3] Vayer, T., Chapel, L., Flamary, R., Tavenard, R., & Courty, N. (2020). Fused Gromov-Wasserstein distance for structured objects. Algorithms, 13(9), 212.
[4] Zhang, Z., Xiang, Y., Wu, L., Xue, B., & Nehorai, A. (2019). Kergm: Kernelized graph matching. Advances in Neural Information Processing Systems, 32.
[5] Xu, H., Luo, D., Zha, H., & Duke, L. C. (2019, May). Gromov-wasserstein learning for graph matching and node embedding. In International conference on machine learning (pp. 6932-6941). PMLR.

**Questions:**

Please answer to the major remarks.

---

### Official Review · Reviewer_Wmbc · 2024-11-01

**Soundness:** 3
**Presentation:** 3
**Contribution:** 2
**Rating:** 5
**Confidence:** 2

**Summary:**

In this paper, the authors propose the Intra-fused Gromov-Wasserstein (IFGW) distance for comparing data with structure and features, such as graphs and point clouds. They point out that the existing method called fused Gromov-Wasserstein has a problem in cross-domain settings due to its approach to directly compare node features between data. As a solution, the proposed method leverages distance matrices of features rather than the features themselves to calculate the distance. The authors also discuss an algorithm for computing IFGW.

**Strengths:**

1. The explanations of previous studies, such as GW distance and fused GW distance, are appropriately provided. Additionally, the authors clearly explain the differences between the existing methods and the proposed method.
2. Quantitative investigations are conducted on clustering tasks.

**Weaknesses:**

1. The definitions of the symbols are insufficient, for example, $\Sigma_m$ in Definition 1, and X in Definition 2.
2. There is a lack of comparison with other methods in the experiments.
3. The experiment in Section 3.3 needs to be conducted with datasets that include more materials. It is not sufficient to provide the distance value for a single material for indicating whether the method is good or not.
4. Although the proposed method claims to have an advantage in cross-domain settings, there are only a few experiments conducted under such conditions.

**Questions:**

1. What is $\Sigma_m$ in Definition 1?
2. What does "which can be simplified as..." right after Definition 2 mean?
3. What is inferred from Figure 2? At the end of page 6, it is stated that "FGW tends to be sensitive to the feature distances between graphs," but the how the figure supports this explanation is unclear.
4. For KMeans clustering in Table 2, what do you use as features of graphs?
5. Are there any results of comparison to other methods in the experiments in Sections 3.2 and 3.3?

---

### Official Review · Reviewer_sUzS · 2024-11-03

**Soundness:** 3
**Presentation:** 3
**Contribution:** 3
**Rating:** 3
**Confidence:** 3

**Summary:**

This paper introduces the Intra-fused Gromov-Wasserstein (IFGW) distance, a novel metric that integrates the properties of both the Wasserstein and Gromov-Wasserstein distances. To address the inherent non-convexity of IFGW, the authors employ entropic regularization as an approach to approximate solutions. The concept of IFGW barycenters is also defined to extend the applicability of this metric.

**Strengths:**

The authors put significant effort into making their presentation comprehensible. In particular, I would like to highlight the following points:

-Table 1.

-The presentation of their new formulation (Equation 9) is visually effective, with the use of color to highlight key comparisons to a preceding formulation (Equation 8).

- In the section "From Distance to Discrepancy," the itemization enhances the understanding of the distinctions between GW, FGW, and the new IFGW, making the comparisons clearer for readers.

**Weaknesses:**

It would be beneficial for the paper to include a more in-depth analytical evaluation of the tool. For example, the authors could provide a theoretical analysis of IFGW's properties as a metric by proving or disproving whether IFGW satisfies the four metric axioms: non-negativity, identity of indiscernibles, symmetry, and the triangle inequality.

**Questions:**

The paper provides a comprehensive overview in Table 1, summarizing related work on graph neural networks and existing transport-based metrics while highlighting their limitations. It would be beneficial if the authors provided clear definitions or examples for the terms "smooth" and "cross-domain" in the context of their work. This addition would help clarify the distinctions made in Table 1.

In the paragraph titled "From Distance to Discrepancy," the authors state: "The GW distance is originally defined on the metric-measure space (mm-space), where C is a strict metric from the structured data, e.g., all-pair shortest path on graphs. However, we can replace the strict structural measure C with a pseudo-metric or semi-metric to generalize the GW distance to GW discrepancy."
This concept is well-known in the literature. Indeed, the GW problem can be formulated in general gauge measure spaces. Therefore, the point made by the authors is unclear to me. It would be beneficial it the authors could clarify the novelty or significance of their approach in light of existing literature on GW in general gauge measure spaces. Additionally, the authors could provide specific examples of how their generalization differs from or improves upon prior work in this area.

---

> ### Comment · Reviewer_sUzS · 2024-11-26
>
> Since I haven't received responces from the authors, I will lower my score.

---

### Official Review · Reviewer_7oxq · 2024-11-04

**Soundness:** 3
**Presentation:** 1
**Contribution:** 2
**Rating:** 3
**Confidence:** 3

**Summary:**

The paper presents a novel similarity metric, the Intra-Fused Gromov-Wasserstein (IFGW) distance, which captures both structural and feature information of graphs. IFGW combines the benefits of Wasserstein and Gromov-Wasserstein distances to provide a more balanced and smooth metric for graph comparison, targeting tasks like graph clustering, classification, and similarity search. This new metric has promising applications in diverse fields, such as bioinformatics, computer vision, and natural language processing.

**Strengths:**

1. The IFGW distance represents a significant advancement in similarity metrics for graph data, integrating both structural and feature information in a single optimal transport framework.
2. This metric has the potential to inspire further research into improved similarity measures for structured data, especially for tasks requiring cross-domain comparisons.

**Weaknesses:**

1. Although generally understandable, the manuscript could benefit from increased clarity and precision, especially in definitions and explanations.
2. The experimental results, while promising, appear only marginally better than existing metrics, leaving questions about the robustness of the improvements.

**Questions:**

1.  In the introduction, the statement regarding “strong performance in graph classification and clustering” should be supported with appropriate citations.
2. In Section 2, terms like “APSP” should include citations or explanations to ensure clarity for readers unfamiliar with the terminology.
3.  Below Equation (2): Clarify which matrix  T  is referenced as “doubly stochastic” to avoid ambiguity. Define “orthonormal domain” more precisely.
4. Clarify the meaning of the  \otimes  symbol in Equation (4)—whether it denotes the tensor product or Kronecker product, or else
5. use a comma at the end of Equation (7).
6. Explain the primary advantages of Equation (13) over Equation (12) in detail, as the improvement is not immediately clear.
7. In discussing the KL projection 14, 15, 17), clarify the difference of uppercase versus lowercase subscripts.
8. Provide definitions for symbols like  \mathcal{T}  in Equation (16) and  D  in the line below Equation (19).
9.  In Table 2, IFGW’s improvement over FGW is relatively minor. This weakens the claim that “IFGW outperforms FGW in clustering tasks across all evaluated datasets in general.”

---

### Note · Authors · 2025-01-22

I have read and agree with the venue's withdrawal policy on behalf of myself and my co-authors.